# S- and N-Co-Doped TiO_2_-Coated Al_2_O_3_ Hollow Fiber Membrane for Photocatalytic Degradation of Gaseous Ammonia

**DOI:** 10.3390/membranes12111101

**Published:** 2022-11-04

**Authors:** Jae Yeon Hwang, Edoardo Magnone, Jeong In Lee, Xuelong Zhuang, Min Chang Shin, Jung Hoon Park

**Affiliations:** Department of Chemical and Biochemical Engineering, Dongguk University, Pildong-ro 1 gil, Jung-gu, Seoul 04620, Korea

**Keywords:** environmental protection, air purification, indoor air pollution, photocatalytic membrane reactor, titanium dioxide, S,N-doped TiO_2_, gaseous ammonia (NH_3_) degradation

## Abstract

This study successfully prepared and tested sulfur- and nitrogen-co-doped TiO_2_-coated α-Al_2_O_3_ (S,N-doped TiO_2_/Al_2_O_3_) hollow fiber (HF) membranes for efficient photocatalytic degradation of gaseous ammonia (NH_3_). Thiourea was used as a sulfur- and nitrogen-doping source to produce a S,N-doped TiO_2_ photocatalyst powder. For comparative purposes, undoped TiO_2_ powder was also synthesized. Through the application of a phase-inversion technique combined with high-temperature sintering, hollow fibers composed of α-Al_2_O_3_ were developed. Undoped TiO_2_ and S,N-doped TiO_2_ photocatalyst powders were coated on the α-Al_2_O_3_ HF surface to obtain undoped TiO_2_/Al_2_O_3_ and S,N-doped TiO_2_/Al_2_O_3_ HF membranes, respectively. All prepared samples were characterized using XRD, TEM, XPS, UV-Vis, SEM, BET, FT-IR, and EDS. S and N dopants were confirmed using XPS and UV-Vis spectra. The crystal phase of the undoped TiO_2_ and S,N-doped TiO_2_ photocatalysts was a pure anatase phase. A portable air purifier photocatalytic filter device was developed and tested for the first time to decrease the amount of indoor NH_3_ pollution under the limits of the lachrymatory threshold. The device, which was made up of 36 S,N-doped TiO_2_/Al_2_O_3_ HF membranes, took only 15–20 min to reduce the level of NH_3_ in a test chamber from 50 ppm to around 5 ppm, confirming the remarkable performance regarding the photocatalytic degradation of gaseous NH_3_.

## 1. Introduction

The human detection threshold for ammonia (NH_3_), a colorless gas with a distinctive odor, is 1.5 ppm, where there is a 20–50 ppm threshold for eye irritation [1]. Indoor air frequently has NH_3_ concentrations that are between 3.5 and 21 times greater than outdoor air, which has a significant impact on air quality [2]. Human emissions, cooking, cleaning, and smoking are only a few of the many indoor sources of NH_3_ [3]. The RD_0_, a self-protective reflex-based indicator, for NH_3_ is nearly equal to the lachrymatory threshold (55 ppm) in humans [1].

Over the past few years, research on indoor NH_3_ has attracted a lot of attention, including efforts to reduce this pollutant and maintain a clean indoor environment. Several studies proposed and developed degradative ways to reduce the concentration of NH_3_, such as using an advanced oxidation process that combines UV irradiation and ozone in livestock buildings [4] or a pilot-scale reactor made up of a set of two-stage biotrickling filters, an influent gas supply system, and a liquid recirculation system [5].

An innovative solution is required to reduce NH_3_ pollution and maintain a clean indoor environment. In order to demonstrate that reducing the gaseous NH_3_ pollutant using a small, portable device has a major positive influence on the environment, it is crucial to investigate a unique photocatalytic membrane reactor.

Recently, more focus has been placed on photocatalysis because of its ability to employ photoenergy to create photoexcited electrons (e^−^) and holes (h^+^), which can stimulate even uphill processes at room temperature [6,7]. Due to its advantages (good chemical stability, low cost, etc.), TiO_2_ has attracted a lot of attention in this cutting-edge field as a photocatalyst, where the anatase TiO_2_ photocatalyst was shown to have more photoactivity than rutile [8,9]. In recent years, it was widely reported that doping TiO_2_ with non-metals, such as sulfur (S) and nitrogen (N), is one of the most efficient strategies to increase its photocatalytic activity under visible light by making TiO_2_’s band gap smaller [10]. The deposition of anatase TiO_2_ photocatalyst thin films on porous ceramic substrates, such as α-Al_2_O_3_-based hollow fiber (HF) membranes, would also be crucial for their widespread use in industrial domains in terms of practical application [11,12].

The purpose of this work was to study the S and N co-doping effects on the NH_3_ photocatalytic degradation capacity (NH_3_ → 0.5N_2_ + 1.5H_2_) of anatase TiO_2_ in a modular prototype photocatalytic membrane reactor with a total of 36 membranes in different light conditions in an indoor atmosphere where NH_3_ can be present with a concentration of 50 ppm. This contribution is the second part of an in-depth study on heterogeneous photocatalytic NH_3_ gas degradation through α-Al_2_O_3_-based hollow fiber membranes functionalized by co-doped TiO_2_ with nonmetallic species, such as nitrogen, sulfur, and carbon [8,9,10,11,12,13,14,15,16,17,18,19,20,21,22,23]. In a previous study, we demonstrated that Al_2_O_3_-based hollow fiber membranes functionalized with nitrogen-doped titanium dioxide (N-TiO_2_) can be used successfully for heterogeneous photocatalytic NH_3_ gas degradation, and then we proposed a novel practical approach to reducing the gaseous NH_3_ pollutant in the environment [11].

In this study, we used a co-doping approach of TiO_2_-coated Al_2_O_3_ HF membranes using S and N ions (S,N-doped TiO_2_/Al_2_O_3_) for the photocatalytic degradation of gaseous NH_3_, and we also studied the performance of a novel modular prototype photocatalytic membrane reactor in a practical NH_3_ concentration. This work aimed to study and propose a new simplified design of a handheld device to reduce the cost of degradation of indoor gaseous NH_3_ by using a portable heterogeneous photocatalytic membrane reactor at room temperature. The ultimate objective of this work was to develop cutting-edge modular designs and innovative experimental procedures for the effective decomposition of NH_3_ in the gas phase, in addition to interesting co-doping combinations from a photocatalytic point of view. In this study, many S,N-doped TiO_2_/Al_2_O_3_ HF membrane-related factors were taken into account during the synthesis and characterization, as well as explored from an application perspective. These achievements can induce a kind of revolution in the design of portable devices and have a great effect on the extensive application of heterogeneous photocatalytic NH_3_ gas degradation in daily life.

## 2. Experimental Materials and Methods

### 2.1. Materials

Titanium (Ⅳ) isopropoxide (TTIP, 97%), thiourea (98%), nitric acid (60%), dimethyl sulfoxide (DMSO, 99.8%), and magnesium oxide were purchased from the Samchun Pure Chemical Co., Ltd. (Seoul, Korea). Tetraethyl orthosilicate (TEOS, 98%) and polyvinylpyrrolidone (PVP, average mol. wt. 40,000, 99.5%) were purchased from Sigma Aldrich (St. Louis, MO, USA). α-Al_2_O_3_ powder (<0.5 μm) was purchased from Kceracell (Daejeon, Korea) and polyethersulfone (PESf) was obtained from Ultrason^®^ (Ludwigshafen, Germany). Without additional purification, all chemicals and reagents were utilized as received.

### 2.2. Synthesis of Undoped TiO_2_ and S,N-Doped TiO_2_ Powders

The undoped TiO_2_ and S,N-doped TiO_2_ powders were synthesized using the sol-gel process. TTIP was utilized as a precursor for titanium [11]. All the information on the procedures used to produce undoped TiO_2_ and S,N-doped TiO_2_ synthesis are shown in the Appendix A. Based on [22], a volume of TTIP (284.22 g/mol) was added to three volumes of deionized water dropwise and blended for an hour. The solution was filtered using a Buchner funnel and a suction filtration unit to remove the solvent quickly, and the remaining water was entirely removed using a 12-h 110 °C drying process. Thiourea (76.12 g/mol) was used as a sulfur and nitrogen source in the synthesis of S,N-doped TiO_2_ powder [13,17,18,23]. As previously documented [22], deionized water was used to dissolve the dopant sources, and the solution was stirred continuously for an hour to make it transparent. For the synthesis of undoped TiO_2_ powder, TTIP was added to the obtained thiourea solution, stirred (one hour), filtered, and then dried (twelve hours, 110 °C). Both of the obtained white solid products were then calcined for three hours at 400 °C in a static air atmosphere. Based on earlier research [22], a calcination temperature of 400 °C was selected to avoid the formation of the rutile phase and then obtain a pure anatase phase with a high specific surface area. The S,N-doped TiO_2_ powders were light yellow after the heat treatment, indicating successful S- and N-co-doping (see Appendix A).

### 2.3. Preparation of α-Al_2_O_3_ Hollow Fiber (HF) Membranes

α-Al_2_O_3_ HF membranes were fabricated using a phase inversion process followed by a thermal treatment [11,12,24]. Briefly, DMSO (33.5 wt%, 78.13 g/mol) and PESf (6 wt%) were mixed for one day. α-Al_2_O_3_ power (59.4 wt%), PVP (0.5 wt%), and 0.6 wt% MgO (40.3 g/mol) were added as sintering aids. The mixture was degassed and extruded under a pressure of five bars and an internal coagulant flow rate of 20 mL/min. The external coagulant surface and the iron nozzle were separated by an air gap of 10 cm. The Al_2_O_3_ green body was placed in deionized water for a day before being dried at 100 °C for twelve hours. The Al_2_O_3_ HF membranes were finally sintered for three hours at 1300 °C in static air. The appropriate membrane length, taking into account the filter assembly procedure, should be around seven centimeters.

### 2.4. Preparation of Undoped TiO_2_ and S,N-Doped TiO_2_/Al_2_O_3_ HF Membranes

Using a dip-coating method with a silica-based binder, photocatalyst powders consisting of undoped TiO_2_ and S,N-doped TiO_2_ were deposited on Al_2_O_3_ HF membranes [25]. The silica binder solution was obtained by adding 2 g of TEOS (208.33 g/mol) to 18 g of deionized water acidified with nitric acid (0.3 g) and stirred for five hours, as reported in previous literature [26]. To prepare the coating solution, 8 g of the prepared silica binder solution was mixed with 78 g of ethanol and 21 g of photocatalytic powder. Under supersonic conditions, acetone was used to clean the ceramic hollow fiber membranes and, when the acetone had completely evaporated, the membranes were placed within the coating solution for one hour. The final treatment consisted of washing the obtained photocatalytic membranes and drying them at 150 °C for twelve hours.

### 2.5. Characterization of Undoped TiO_2_ and S,N-Doped TiO_2_ Powder and Photocatalytic Membranes

X-ray diffraction (XRD, Ultima IV, Rigaku, Tokyo, Japan) using Cu Kα radiation from 20° to 80° in the 2θ range at room temperature, transmission electron microscopy (TEM, JEM-F200, JEOL Ltd., Tokyo, Japan), X-ray photoelectron spectroscopy (XPS, Veresprobe II, ULVAC-PHI, Chigasaki, Japan) within a range of 0–1200 eV, UV-Vis diffuse reflectance spectrophotometry (UV-Vis DRS, SolidSpec-3700, Shimadzu, Kyoto, Japan) within a range of 200–800 nm, scanning electron microscopy (SEM, model S-4800, Hitachi, Tokyo, Japan) with an energy dispersive spectrometer (EDS, model S-4800, Hitachi, Tokyo, Japan), photoluminescence spectroscopy (PL, LabRAM HR-800, Horiba Ltd., Kyoto, Japan) within a range of 400–800 nm, and Fourier-transform infrared spectroscopy (FT-IR, Spectrum Two, Perkin Elmer, Waltham, MA, USA) within a range of 500–4000 cm^−1^ were employed to characterize the obtained products and membranes. The average crystallite size was calculated from the full width at half maximum of the diffraction peaks by using the Debye–Scherrer equation [27]. The Brunauer–Emmet–Teller (BET) method was used to quantify the specific surface area using N_2_ adsorption–desorption measurements at 77 K (ASAP2020, Micromeritics, Atlanta, GA, USA). Further details are reported in our previous study [26].

### 2.6. Photocatalytic Degradation of Gaseous Ammonia

Before testing the NH_3_ decomposition performance, the prepared photocatalytic membranes were assembled into a module. Figure 1a shows the basic concepts of the photocatalytic filter-type module. The shape of the filter-type module was a cylinder (20 cm × 10 cm). Thirty-six photocatalytic membranes were inserted into the corresponding holes of an acrylonitrile butadiene styrene (ABS) resin structure in the form of a disk (see Figure 1b). It was decided to use a 36-membrane configuration because it offered a reasonable balance between robustness, cost, and performance [11]. Each membrane was fixed and sealed using epoxy resin as adhesive and the side part of the filter-type module was covered using transparent protection. The effective single membrane area was calculated to be approximately 0.01 m^2^. At the axis of the cylindrical filter-type module, a hole was made to leave space to install a triangular LED light source to irradiate the photocatalytic membranes from inside the filter-type module (see Figure 1c). The NH_3_ mixed with air (99.999%) was supplied and calibrated using mass flow controllers. The NH_3_ concentration was analyzed using gas chromatography (GC).

This small photocatalytic filter-type device was installed in a simulated indoor environment, such as a closed room, and studied using two types of reactor: one to evaluate the “photocatalytic filter-type module” (reactor #1) in extreme conditions (500 ppm NH_3_) and another one to evaluate the “photocatalytic air purifier” performance in a real situation (50 ppm NH_3_) [1,2].

#### 2.6.1. Photocatalytic Filter-type Module Evaluation (Reactor #1)

To evaluate the performance of a photocatalytic filter-type module, the portable device was placed in a hermetically sealed test chamber that was completely isolated from the outside. The modular device was equipped with a DC fan and a DC power supply. The test chamber was, in turn, closed in a dark room to avoid any external interference. Figure 2 shows the schematic image of the photocatalytic filter-type module. Initially, the NH_3_ was supplied continuously through reactor #1 as a common flow-type or flow-through reactor until stabilization of the entire system was achieved. When the concentration of NH_3_ stabilized, the LED light source was turned on. At the same time, the system was switched from the initial flow-type reactor to a batch-type reactor operating in an unsteady state for 30 min.

#### 2.6.2. Photocatalytic Air Purifier Evaluation (Reactor #2)

The photocatalytic filter-type module was reassembled into an acrylic structure to produce a portable air purifier device with a more compact size. The air-purifying efficiency in the contained environment was tested at room temperature with an NH_3_ concentration of 50 ppm. Figure 3 shows a schematic image the reactor #2 used for the evaluation of photocatalytic air purifier performance.

There were three steps to testing reactor #2. Figure 4 shows the three steps for the evaluation of photocatalytic air purifier performance. The three steps were stabilization, irradiation, and evaluation. The stabilization process was the same as the previous experimental system (see reactor #1). Once the NH_3_ concentration was stabilized, the valves at the top and bottom of the (acrylic) square-type dark-box chamber were switched to capture and then trap the NH_3_ in the test chamber and the irradiation was turned on. This step was the simulation of a possible situation in an indoor environment without air circulation under a blue LED light source. The test chamber was once again opened in the final evaluation step to examine the residual NH_3_ and to check whether the concentration of NH_3_ was under the limits of the lachrymatory threshold (50 ppm) in humans [1].

## 3. Results and Discussion

### 3.1. Chemical and Physical Analysis of the Produced Samples

Figure 5 shows the XRD patterns of the undoped TiO_2_ and S,N-doped TiO_2_ photocatalyst powders. As can be seen, the X-ray diffraction pattern for both powders had large peaks, indicating small crystallites. By comparing with the JCPDS card information, it can be seen that the crystal phase of both samples was a pure anatase phase with a peak at 25 degrees of 2θ corresponding to (101) plane diffraction of the anatase phase of TiO_2_ (JCPDS card number 21-1272).

The unit cell parameters for the undoped TiO_2_ and S,N-doped TiO_2_ photocatalyst powders are given in the Appendix A (see Appendix A). Anatase TiO_2_ has a tetragonal symmetry class (space group: I41/amd; space group number: 141) and its lattice constants *a*, *b*, and *c* are 3.7848 Å, 3.7848 Å, and 9.5124 Å, respectively [28]. The lattice constants *a*, *b*, and *c* for the prepared undoped TiO_2_ photocatalyst powder prepared in this work were 3.7849 Å, 3.7849 Å, and 9.4883 Å, respectively. The calculated unit cell volume for undoped TiO_2_ photocatalyst powders (unit cell volume = 135.92 Å^3^) was very close to standard TiO_2_, which has a unit cell volume of 136.27 Å^3^ [28]. The lattice parameters of the prepared S,N-doped TiO_2_ photocatalyst powder were as follows: *a* = *b* = 3.7916 Å, and *c* = 9.5055 Å (unit cell volume = 136.65 Å^3^). These parameters corresponded to the experimental values (*a* = *b* = 3.790 Å and *c* = 9.509 Å) for nano-N-doped TiO_2_ powders synthesized using solvothermal processes [29], which indicated that the crystal structure of the TiO_2_ photocatalyst powder was changed with the co-doping of TiO_2_ with S and N ions.

In addition, the crystallite size of both synthesized photocatalyst powders was determined via XRD using the high-intensity peak at a low angle (2θ ≅ 25°) with the Debye–Scherrer formula [27] as follows: $$D=\frac{K\lambda }{\beta_{r}\;cos\theta }$$
where *D*, *K*, *λ*, *β_r_*, and *θ* are the average crystallite size, shape factor, X-ray wavelength, full width at half maximum of the diffracted peak (FWHM), and Bragg’s angle, respectively. The crystallite sizes of the undoped TiO_2_ and S,N-doped TiO_2_ photocatalyst powders were estimated to be 6.90 nm and 6.77 nm, respectively. In accordance with previously reported data [30,31,32], the doping approaches resulted in a decrease in the average crystallite size of photocatalyst powders.

The BET surface area of each photocatalyst powder was also determined using N_2_ adsorption–desorption measurements. The BET surface area of undoped TiO_2_ was equal to ~50 m^2^/g. Doping S- and N- ions into TiO_2_ led to an increase in the surface area to ~91 m^2^/g. Based on previous experience, the surface area of S-doped TiO_2_ prepared through a calcination conversion route was calculated to be about 254.4 m^2^/g, which is larger than the surface area of undoped TiO_2_ (77.1 m^2^/g) [33]. In contrast, it was found that the surface area of TiO_2_ material synthesized using simple co-precipitation decreased with nitrogen doping [34]. In particular, this apparent contradiction is a problem that was highlighted recently by Piątkowska et al. [10], who noted that the effect of the co-doping on the specific surface area of TiO_2_ material was strongly dependent on the synthesis method.

The microstructures of the undoped TiO_2_ and S,N-doped TiO_2_ photocatalyst powders were examined using TEM analysis at Figure 6. Previous research [19,20] found that the undoped TiO_2_ was organized into monodispersed spherical TiO_2_ particles with an average diameter of approximately 10–12 nm, as is also shown in Figure 6a,b. Figure 6c,d shows that spherical and irregular-shaped TiO_2_ particles were mixed in the S,N-TiO_2_ photocatalyst powder. Irregular-shaped TiO_2_ particles are shown in Figure 6d.

Figure 7 shows the XPS survey spectra of the undoped TiO_2_ and S,N-doped TiO_2_ powders. The undoped TiO_2_ contained only Ti, O, and C elements, with sharp photoelectron peaks appearing at the Ti2p, O1s, and C1s binding energies (see Figure 7a). Figure 7b shows the XPS survey spectra of the undoped TiO_2_ powders. Figure 7c–f shows high-resolution XPS spectra of the Ti2p, O1s, S2p, and N1s regions, respectively. According to the XPS survey spectrum, the peak of S2p around 167.0 eV was found to have moved negatively in the XPS survey spectrum compared with the sulfur’s typical binding energy in pure SO_4_^2−^ (169.0 eV) [10,35]. The peak at about 400 eV in the S,N-doped TiO_2_ powder was presumed to be produced by the O-Ti-N structural bond [11,23]. In conclusion, it was confirmed that the synthesized S,N-doped TiO_2_ powder contained not only sulfur ions but also nitrogen ions with characteristic peaks appearing at the binding energies of 167.0 eV (S2p) and 399 eV (N1s). In comparison with the undoped TiO_2_ sample, it was noted that for the S,N-doped TiO_2_ sample, the Ti2p and O1s peak shifted to lower binding energies. This result suggested the presence of S,N-Ti-O and S,N-O bonds, which supported the fact that the sulfur and nitrogen ions were successfully co-doped into the TiO_2_ crystalline structure [21]. In addition, it is interesting to note here that the peaks for C-C bonds (285.2 eV) and O-C=O bonds (290 eV) were typically attributed to carbon that was adsorbed on the surface of the photocatalyst as a contaminant, indicating that the C element was not doped into the TiO_2_ lattice [36,37]. As reported in previous work [26], it can be hypothesized that a small amount of silicon dioxide (SiO_2_) derived from the silica binder solution (TEOS) is present in both undoped TiO_2_ and S,N-doped TiO_2_/Al_2_O_3_ HF membranes, whereas, eventually, residual SiO_2_ has no photocatalytic activity [38] (see Appendix A).

Figure 8a shows the absorbance properties of undoped TiO_2_ and S,N-doped TiO_2_ photocatalyst powders between 200 nm and 800 nm. The presence of the sulfur and nitrogen ions in the TiO_2_ lattice extended the absorbance from the UV region, with the absorption edge at about 400 nm, to the visible light region (400~600 nm). These results are in good agreement with data obtained by other authors [20,39,40,41].

By converting the UV-Vis diffusive reflectance spectra data into a Tauc plot, the band gap energy can be known by measuring the cut-off wavelength. The calculated band gap energies of the samples are shown in Figure 8b. The band gap energies for the undoped TiO_2_ and S,N-doped TiO_2_ photocatalysts were 3.17 eV and 2.42 eV, respectively. According to the band gap energies found in this study, co-doping TiO_2_ with S and N ions caused the band gap to decrease. In comparison to undoped TiO_2_, the band gap for N,S-doped TiO_2_ moved to a higher negative potential, signifying increased electron injection [18]. These observations support the hypothesis that S,N co-doping can reduce the gap energy and move TiO_2_’s photocatalytic activity into the visible spectral region [18].

To understand the behavior of holes and light-generated electrons, the photoluminescence (PL) emissions of the undoped TiO_2_ and S,N-doped TiO_2_ photocatalyst powders were also examined (see Appendix A) at wavelengths ranging from 400 to 800 nm. The emission peaks for the undoped TiO_2_ and S,N-doped TiO_2_ photocatalyst powders were 585 nm and 530 nm, respectively. The emission spectra of the two photocatalysts had similar shapes. According to previous studies, an N-doped TiO_2_ photocatalyst has a lower PL intensity than an undoped TiO_2_ sample [11,42].

Figure 9a shows the cross-section SEM image of the S,N-doped TiO_2_/Al_2_O_3_ HF membrane. As shown in the SEM images in Figure 9a, the Al_2_O_3_ HF membrane prepared using the phase inversion technique had a combination of a finger-like structure in the inner and outer regions and a sponge-like structure in the middle. The wall thickness of the Al_2_O_3_ HF membrane was about 430 μm. For additional SEM images, see Appendix A. From the SEM images of the cross-section of the S,N-doped TiO_2_/Al_2_O_3_ HF membrane with different magnifications, the S,N-doped TiO_2_ photocatalytic layer was visually estimated as being 6–7 nm thick.

We used an EDS line-scanning measurement on the S,N-doped TiO_2_/Al_2_O_3_ HF membrane to further investigate the sample’s structure. The S,N-doped TiO_2_/Al_2_O_3_ HF membrane’s EDS analysis result is shown in Figure 9b, where the scanning spectra of Ti, O, and Al elements are presented in different colors. The Ti signals emerged simultaneously on both the S,N-doped TiO_2_/Al_2_O_3_ HF membrane sides, while the Al signals first appeared in the middle of the membrane. In addition, we estimated the thickness of the photocatalytic layer to be between 8 and 10 nm based on the intensity of the Ti signal. Although the S and N signals were difficult to obtain due to the low concentration of these elements on the photocatalytic layer, the deposition of S,N-doped TiO_2_ was clearly indicated.

The elemental distributions of the corresponding C, O, Al, Ti, and S elements on the surface of the S,N-doped TiO_2_/Al_2_O_3_ HF membrane were examined using the EDS mapping analysis. As shown in Figure 10, Ti and S elements were found, whereas the element N was missed due to its low content. The elemental distributions of S and N were 3.2 and 1.6 weight percent, respectively (see Appendix A).

### 3.2. Photocatalytic Degradation of Gaseous Ammonia

#### 3.2.1. NH_3_ Gas Removal Using the Photocatalytic Filter-type Module (Reactor #1)

The photocatalytic degradation of gaseous NH_3_ was investigated under different light conditions in reactor #1 (see Figure 2) at room temperature. Figure 11 shows the result of the photocatalytic elimination of NH_3_ using the photocatalytic filter-type module. Figure 11a illustrates the photocatalytic degradation of gaseous NH_3_ using an undoped TiO_2_/Al_2_O_3_ HF membrane. When tested under a UV light source, a photocatalytic filter-type module composed of 36 undoped TiO_2_/Al_2_O_3_ HF membranes showed good photocatalytic activity, with the initial concentration of gaseous NH_3_ being reduced to zero after approximately 25–30 min. By contrast, the photocatalytic ability seemed very low under white and blue LED light sources. Under these conditions, a decrease of between 20 and 30% in gaseous NH_3_ was observed in reactor #1 with the TiO_2_/Al_2_O_3_ HF membranes.

The behavior of the photocatalytic filter-type module composed of 36 S,N-doped TiO_2_/Al_2_O_3_ HF membranes was studied using the same conditions (reactor #1) and light sources (white, blue, and UV) as the undoped TiO_2_ system. The NH_3_ photocatalytic degradation performance of the S,N-doped TiO_2_/Al_2_O_3_ HF membranes in reactor #1 is shown in Figure 11b. The NH_3_ photocatalytic degradation capabilities of the S,N-doped TiO_2_/Al_2_O_3_ HF membranes were superior to those in the undoped scenario, as expected, independent of the light sources. In particular, after only 15 min of exposure to a UV LED light source, the photocatalytic filter-type module made with the S,N-doped TiO_2_/Al_2_O_3_ HF membranes had a maximum NH_3_ photocatalytic degradation capability of 100%.

Furthermore, the photocatalytic activity of the S,N-doped TiO_2_/Al_2_O_3_ HF membranes under blue LED light (Figure 11b) was comparable to that of undoped TiO_2_/Al_2_O_3_ HF membranes under UV LED light (Figure 11a). This result showed that S- and N-co-doping is a useful strategy for enhancing photocatalytic activity when using a blue LED light source.

The photoformed hole on the S,N-doped TiO_2_/Al_2_O_3_ HF membrane oxidizes NH_3_ to produce a proton and an amide radical (•NH_2_) as part of the ammonia decomposition mechanism (NH_3_ → 0.5N_2_ + 1.5H_2_). The •NH_2_ then produces hydrazine (N_2_H_4_), which can be further decomposed to produce N_2_ and H_2_ [6,7,10]. The S,N-doped TiO_2_/Al_2_O_3_ HF membranes’ increased UV light photocatalytic activity over undoped TiO_2_/Al_2_O_3_ HF membranes was attributed to their small band gap and visible light absorption.

#### 3.2.2. NH_3_ Gas Removal Using the Photocatalytic Air purifier (Reactor #2)

The NH_3_ gas removal capacity of a compact photocatalytic air purifier consisting of S,N-doped Al_2_O_3_ HF membranes was investigated under a blue LED light source in a test chamber (reactor #2) that simulated an indoor environment of gaseous NH_3_ without air circulation. This experimental protocol had two objectives: (a) to verify the proposed photocatalytic air purifier in a test chamber that replicated the environment of indoor NH_3_ gas pollution, and (b) to lower the NH_3_ gas pollution below levels that are unhealthy and cause eye irritation in humans [1,2]. Figure 12 presents the results of this experiment. In particular, Figure 12 presents the variations from the initial NH_3_ indoor concentration (50 ppm) as a function of time for the three considered steps (stabilization, irradiation, and evaluation).

In the first step, where NH_3_ was continually delivered through reactor #2, the test chamber had to be stabilized for an hour. The solid line in Figure 12 represents the linear extrapolation of the data collected during the stabilization of the system and it serves as a visual representation of this step. During the second step of the process, called “irradiation”, the in and out valves were closed and, simultaneously, the blue LED light source was switched to simulate indoor gas pollution of gaseous NH_3_ without air circulation. It was noted that the NH_3_ concentration seemed to gradually decrease due to the residual gas in the line. When the test chamber was finally opened during the evaluation step, the NH_3_ concentration appeared to be very low in contrast to the initial concentration (50 ppm). The amount of remaining NH_3_ inside the test chamber was approximately 5 ppm. The detected NH_3_ concentrations in the test chamber were below the tolerated limits specified in the literature [1,2]. This result is significant because it supports the benefits of the proposed approach for the gaseous NH_3_ photocatalytic degradation reaction in terms of protecting the environment and public health.

## 4. Conclusions

In recent years, ammonia (NH_3_) has been a significant contributor to indoor air pollution. In this study, we proposed that an advanced, small, and transportable photocatalytic air purifier made with S,N-doped TiO_2_/Al_2_O_3_ hollow fiber (HF) membranes is a good candidate for the photocatalytic degradation reaction of indoor NH_3_ gas pollution.

HFs made of α-Al_2_O_3_ were produced by using a phase-inversion approach along with high-temperature sintering. Titanium (IV) isopropoxide (TTIP) and thiourea were used as titanium and sulfur-nitrogen-doping sources, respectively, to produce S,N-doped TiO_2_ photocatalyst powder. An undoped TiO_2_-based photocatalyst was also synthesized and then studied as a reference. An 8–10 nm layer of undoped TiO_2_ or S,N-doped TiO_2_ was successfully deposited on Al_2_O_3_ HF membranes.

The anatase structure was observed in all samples. The lattice cells confirmed the effective doping of S and N ions into the lattice of TiO_2_ photocatalysts. The surface area of the undoped TiO_2_ was slightly smaller than the S,N-doped TiO_2_. In addition, the co-doping of TiO_2_ with S and N ions was also demonstrated through XPS, UV-Vis, and PL emissions. The S and N co-doping strategy was found to be advantageous in optimizing the photocatalytic degradation reaction of indoor NH_3_ gas pollution. S,N-doped TiO_2_ coated on Al_2_O_3_ HF membranes exhibited excellent performance for NH_3_ decomposition under UV light. When using the novel photocatalytic air purifier device made up of 36 S,N-doped TiO_2_/Al_2_O_3_ HF membranes, it took only 15–20 min to reduce the level of NH_3_ from 50 ppm to around 5 ppm in a test chamber at room temperature.

## Figures and Tables

**Figure 1 membranes-12-01101-f001:**
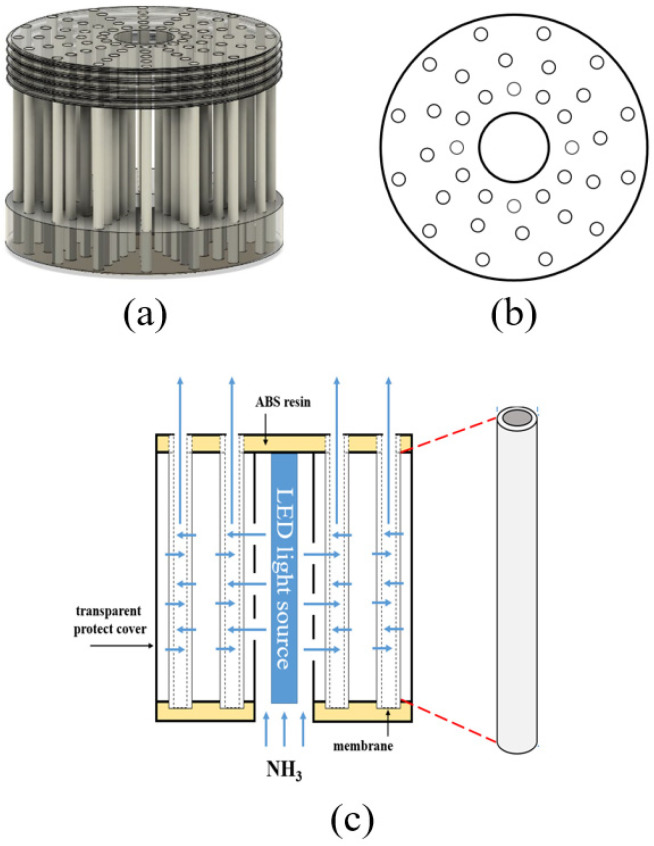
(**a**) Schematic diagram of the photocatalytic filter-type module with (**b**) 36 membranes. (**c**) The arrangement of the LED light source in the middle of the photocatalytic filter-type module.

**Figure 2 membranes-12-01101-f002:**
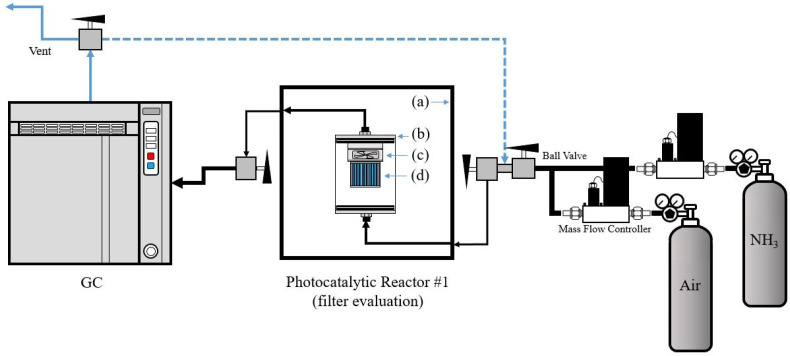
Schematic image of reactor #1 for the evaluation of the photocatalytic filter-type module performance: (**a**) dark box, (**b**) cylindrical test chamber, (**c**) DC fan and DC power supply, and (**d**) photocatalytic filter-type module.

**Figure 3 membranes-12-01101-f003:**
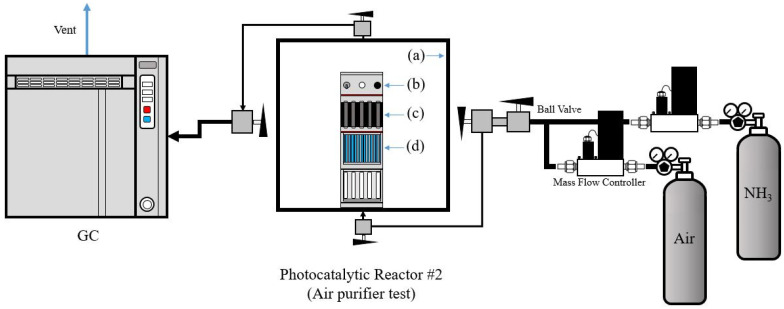
Schematic image of reactor #2 for the evaluation of photocatalytic air purifier performance in a simulated indoor environment without air circulation: (**a**) (acrylic) square-type dark-box chamber, (**b**) central control system for the DC fan and blue LED light source, (**c**) DC fan and DC power supply, and (**d**) photocatalytic air purifier.

**Figure 4 membranes-12-01101-f004:**
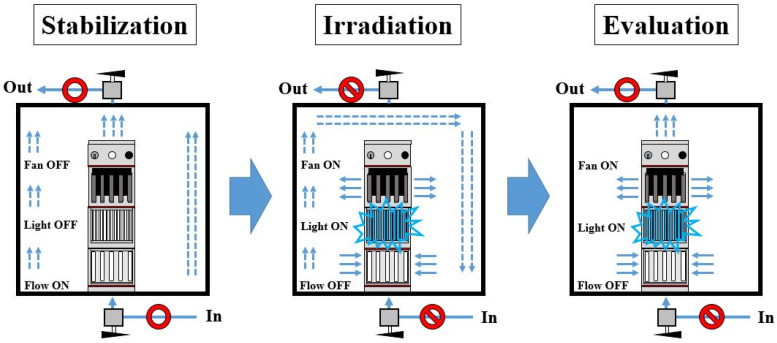
Schematic image of the three-step process for the evaluation of the photocatalytic air purifier performance (reactor #2) with a simulation of a real case of an indoor environment without air circulation.

**Figure 5 membranes-12-01101-f005:**
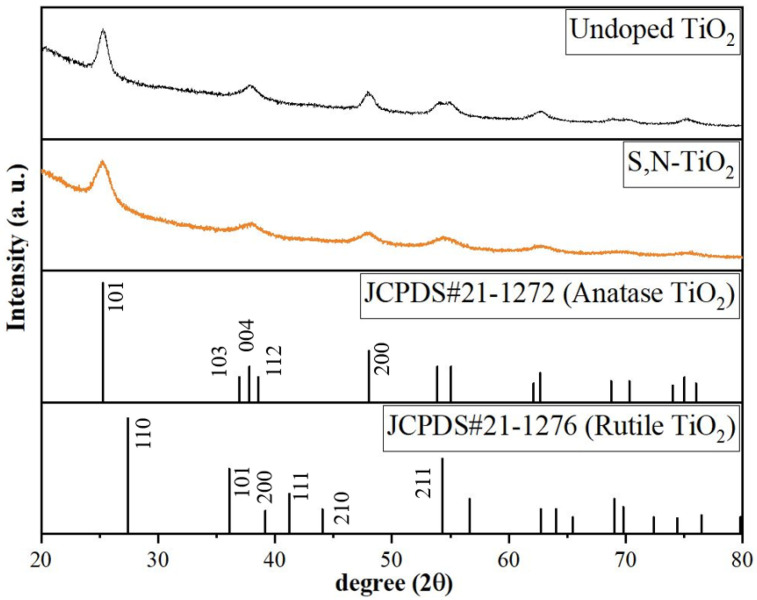
XRD patterns of undoped TiO_2_ and codoped S,N-doped TiO_2_ photocatalyst powders. The XRD pattern of the anatase phase of TiO_2_ (JCPDS card number 21-1272) and the rutile phase of TiO_2_ (JCPDS card number 21-1276) are also provided for comparison purposes (bottom patterns).

**Figure 6 membranes-12-01101-f006:**
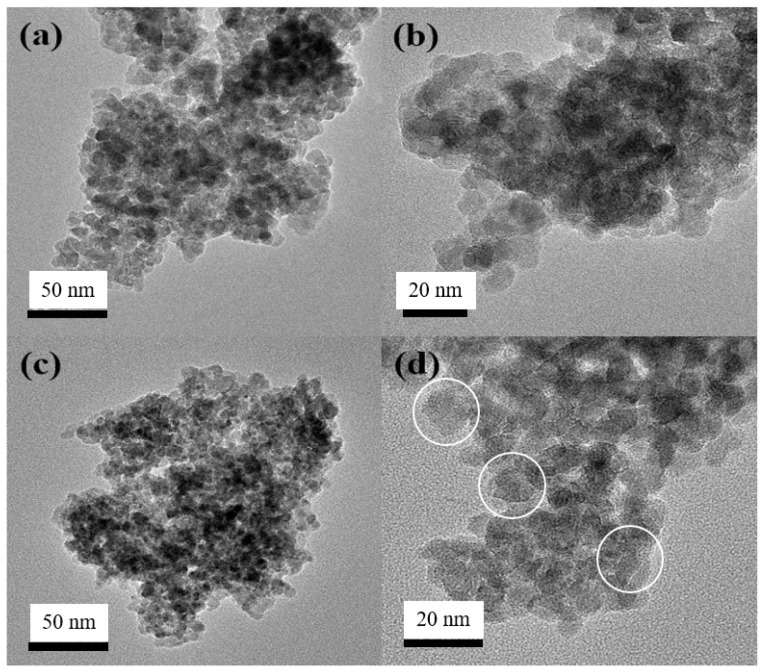
TEM images of (**a**,**b**) undoped TiO_2_ samples and (**c**,**d**) S,N-doped TiO_2_ samples.

**Figure 7 membranes-12-01101-f007:**
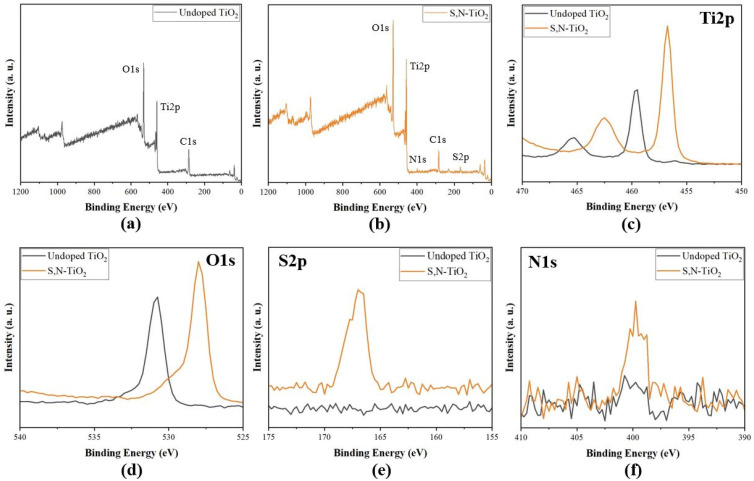
XPS survey spectra of (**a**) undoped TiO_2_ and (**b**) S,N-doped TiO_2_. High-resolution XPS spectra of the (**c**) Ti2p, (**d**) O1s, (**e**) S2p, and (**f**) N1s regions.

**Figure 8 membranes-12-01101-f008:**
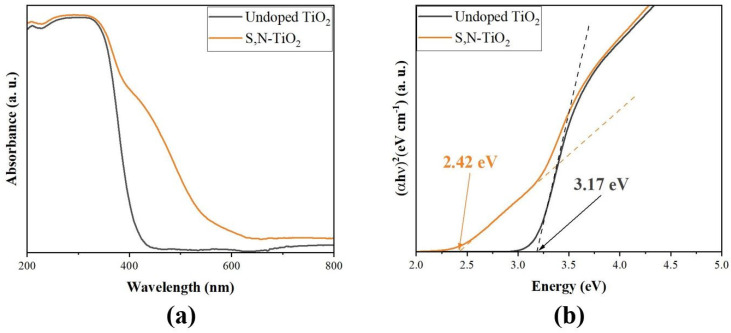
(**a**) UV-Vis spectra and (**b**) Tauc plot of undoped TiO_2_ and S,N-doped TiO_2_ photocatalyst powders.

**Figure 9 membranes-12-01101-f009:**
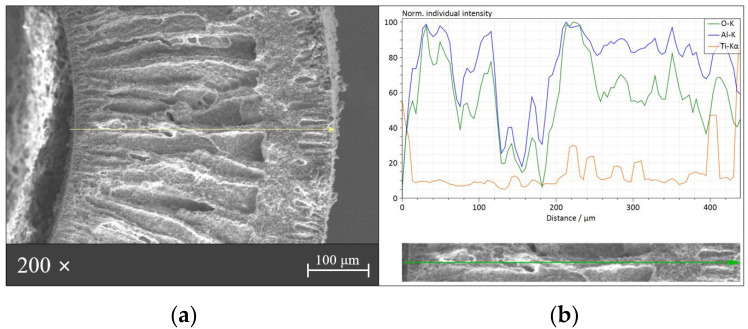
(**a**) SEM image and (**b**) element distribution spectra obtained via EDS line-scanning analysis along the S,N-doped TiO_2_/Al_2_O_3_ HF membrane.

**Figure 10 membranes-12-01101-f010:**
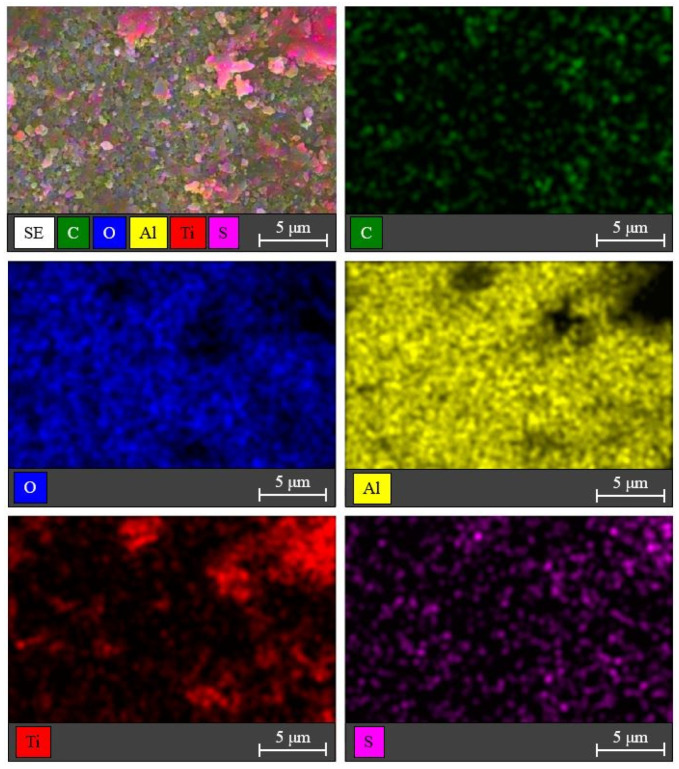
SEM photos and EDS mapping analysis (C, O, Al, Ti, and S elements) for the S,N-doped TiO_2_/Al_2_O_3_ HF membrane surface.

**Figure 11 membranes-12-01101-f011:**
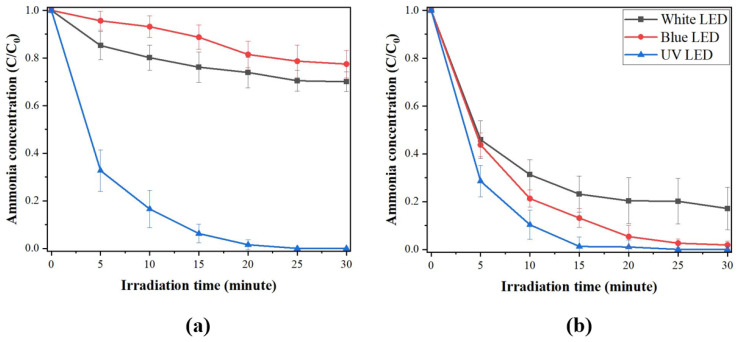
NH_3_ photocatalytic degradation capacity of (**a**) TiO_2_/Al_2_O_3_ HF membrane and (**b**) S,N-doped TiO_2_/Al_2_O_3_ HF membrane under white, blue, and UV LED light sources (reactor #1).

**Figure 12 membranes-12-01101-f012:**
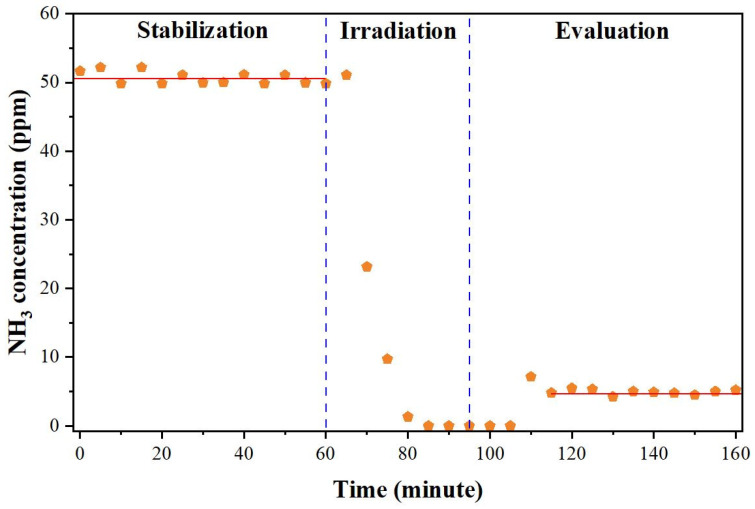
NH_3_ gas removal capacity of the photocatalytic air purifier composed of 36 S,N-doped TiO_2_/Al_2_O_3_ HF membranes in a test chamber without air circulation under a blue LED light source for 30 min (reactor #2).

## Data Availability

Not applicable.

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
