# Peer review of "S- and N-Co-Doped TiO2-Coated Al2O3 Hollow Fiber Membrane for Photocatalytic Degradation of Gaseous Ammonia"

_membranes, 2022, doi:10.3390/membranes12111101_

Round 1

Reviewer 1 Report

Dear Editor

Thank you for the invitation to review this review paper termed as:
Ref: membranes-1972740

Title: S- and N-codoped TiO2-coated Al2O3 hollow fiber membrane for photocatalytic degradation of gaseous ammonia

Authors have reviewed the syntheses, characterizations and applications of un-doped and S, N doped TiO2 nanoparticles as air purifier system for removal of Ammonia gas.

The selected subject for the manuscript is interesting, but the author's information for syntheses of materials is not sufficient. Some point as are follows:

1- The molecular weights of polymeric materials are not reported. Please exactly identify the used materials.

2- The preparation procedure did not meet the good quality. The authors do not mention the exact used procedure.

3- Also the experimental section is questionable. If you add the TTIP to water very simply, which parameters affects to the control of the shape and size of prepared nanoparticles???

4- You know the thiourea is an organic matter. It is simply absorb onto the surface of polar solids. When thiourea was heated at air atmosphere, it decomposed to SOx and NOxs. How could you conclude the insertion of ionic sulfur and nitrogen structures in nanoparticle lattice???

5- …

On the other hand, for decoration of modified and unmodified TiO2 nano-catalyst to the hollow fiber, the author used silica nano or micro structures. How the author confirmed that the surface of nano-catalyst did not coat by silica sol?

 I am not satisfied with this work, and I could not recommend this work for publication in membranes. I think this work have weaknesses in preparation and characterization parts and could not show the strength of the selected subject.

 Regards

Reviewer 2 Report

The paper is focused on S- and N-codoped TiO2-coated Al2O3 hollow fiber membrane  for photocatalytic degradation of gaseous ammonia.

The subject of this manuscript is very similar to the work  published in this journal: Membranes 202212(7), 693; https://doi.org/10.3390/membranes12070693.

The authors should discuss this paper and compare the results obtained in this work with the use of Al2O3-based hollow fiber membranes functionalized by N-TiO2 to remove NH3 pollutants.

In Section, photocatalytic degradation of gaseous ammonia, the authors report that “the shape of the filter-type module is a compact cylinder and the photocatalytic membranes are inserted into the holes of a  disk-shaped acrylonitrile butadiene styrene (ABS) resin structure designed to allow 36 membranes in a single filter-type module”. Why did use 36 membranes  in this experiment?

Please, put the letters (a), (b) and (c), in Figure 1, in according to description of the legend.

In  description of the Fig. 5, please change X-ray diffraction spectra by X-ray diffraction pattern.

In Fig. 5 discussion the author report that the XRD patterns of undoped TiO2 and S,N-doped TiO2 photocatalyst  powders have large peaks indicating small crystallites. The larges peaks indicate also that these powders have amorphous character. This behavior can be due the low calcination temperature, at 400°C. The authors should show the evolution of crystallinity as a function of temperature, to prove that the powders obtained are related to the  anatase phase of TiO2.

The authors report that the “crystallite size for both synthesized photocatalyst powders were estimated to be around 6.90 nm and 6.77 nm, respectively, based on the XRD results”. Please, the method used to calculate the average crystallite size must be reported.

The powder morphology of the photocatalysts must be determined using the BET analysis.

Reviewer 3 Report

1.       The introduction part is too cumbersome to highlight the innovative point and the main research content, which needs substantial modification? Also discuss TiO2 and Al2O3 in one paragraph in detail. The novelty of the research must be emphasized.

2.   Giving a detail characterization section in paper that which machines and analysis were used for as-prepared materials.

3.       It’s is better to provide the digital photograph of samples to prove the color of samples.

4.       How to optimize the conditions for the synthesis of photocatalyst (as showed in the Experimental section)?

5.       The Photoluminescence spectra (Pl) of materials need to provide the actual red shifting upon modification. Also, the SEM morphology is not enough to elevate the texture.

6.       Many analyses in photocatalysis play a vital role and therefore need to provide Brunauer-Emmett-Teller (BET), Barrett-Joyner-Halenda (BJH) and Fourier transform infrared spectroscopy (FTIR) of samples.

7.       Provide the C/N molar ratio and elemental composition (EPR) of catalysts.

8.       Analysis results should be compatible with each other and support each other.

9.       Part photocatalytic mechanism for all these photocatalytic degradation portion. Should be written in more detail, clearly describing the reaction mechanism. why the superior sample improve the photocatalytic activity? How the charge separation was improved, or it is only due to the increasing of surface area. The authors should compare the redox potentials of H+/H2 and O2/O2·- with the conduction band, the redox potentials of ·OH/H2O with the valance band of superior sample, to verify the possibilities of these reaction will be mentioned at the end of paper.

10.   The abstract and conclusion section should be rewritten, clearly mentioning the key findings of the work with numerical values. (Conclusion you written is a joke with this paper and science)

11.   The English of this manuscript needs to be improved, such as making every sentence shorter and correct.

12.   Reference list must follow the journal style.

13.   There are many spaces in paper but in short giving a good magnification to all figures. All figure is blurry and low magnification. Please improve my all comments given on your paper, otherwise in revision will be reject from my hands.

Reviewer 4 Report

The authors reported the work on "S- and N-codoped TiO2-coated Al2O3 hollow fiber membrane for photocatalytic degradation of gaseous ammonia" very well. Material has been synthesized and characterized by various techniques nicely. Materials used in this current work for efficient photocatalytic degradation of gaseous NH3.The contents of the paper are very useful to the scientific society. Some points should be explained before publication:

1. Novelty of the research work should be clearly mentioned in the manuscript.

2. The band gap energies for undoped  TiO2 and codoped S, N-TiO2 photocatalyst were 3.17 eV and 2.42 eV, respectively. Why? Give a proper explanation.

3. Compare your data with published research work.

4. English language should be checked.

Round 2

Reviewer 1 Report

Many thanks for your valuable efforts in revising the manuscript.

Reviewer 2 Report

The suggestions  and recommendations were considered by authors and the manuscript was modified accordingly.

Reviewer 3 Report

All queries arise by me has been well solved and reflecting a good attraction for future science and researchers.